# Potential of Naftifine Application for Transungual Delivery

**DOI:** 10.3390/molecules25133043

**Published:** 2020-07-03

**Authors:** Indrė Šveikauskaitė, Vitalis Briedis

**Affiliations:** 1Department of Clinical Pharmacy, Lithuanian University of Health Sciences, Sukilėlių pr. 13, Kaunas 50161, Lithuania; vitalis.briedis@lsmuni.lt; 2Institute of Pharmaceutical Technologies, Lithuanian University of Health Sciences, Sukilėlių pr. 13, Kaunas 50161, Lithuania

**Keywords:** transungual delivery, polymeric nail lacquers, naftifine, chemical enhancement, fractional laser, onychomycosis

## Abstract

Naftifine is used to treat fungal skin infections as it inhibits dermatophytes, which are the cause of onychomycosis. However, naftifine’s ability to permeate the human nail barrier has not been investigated, thus, the antimycotic potential is not clearly established. This work aims to evaluate the effect of penetration enhancing factors on the accumulation of naftifine hydrochloride through human nail clippings. Naftifine polymeric nail lacquers with Eudragit RL100 were developed as a suitable delivery system. Low penetration of naftifine into nail has been determined as less than 10% of applied drug dose accumulated in the nail layers. Incorporation of thioglycolic acid into formulations resulted in increased accumulation of antifungal agent in the nail layers by 100% compared with a control group. Salicylic acid did not effect naftifine accumulation in the human nail. The permeation of naftifine through the nail increased by threefold when the thioglycolic acid-containing formulation was applied and the nail was pretreated with a fractional CO_2_ laser. Structural changes of the nail barrier, induced by fractional CO_2_ laser, were visualized by microscopy. The results suggest, that naftifine nail penetration could be significantly increased when physical and chemical enhancing factors are applied.

## 1. Introduction

The structure and specificity of biological barriers affect the penetration and bioavailability of drug substances. It is important to use reliable and reproducible models that can reproduce the properties of natural barriers for reliable characterization of topical preparations. Changes of barrier properties in pathology become an additional challenge in permeation studies, as the human nail is one of the natural barriers that changes during infections, and creates additional limitations for efficient therapy.

Onychomycosis (nail fungus)—a fungal contagious disease of the nail bed or nail plate, accounting for about 50% of all cases of nail diseases globally [1]. Oral treatment is the most effective, but long-term (at least 6 months) drug administration may lead to severe adverse reactions, interactions with other medicines and is not acceptable during lactation or pregnancy [2]. Local treatment of onychomycosis lasts over a year, with the efficacy of up to 50%, but interactions with other medicinal products are avoided, and treatment can be given to all patient groups [3]. However, local infection treatment is complicated by the structure of the nail, made up of 80% keratin. Keratin molecules are linked by disulfide bonds, resulting in one of the strongest barriers in the human body, which also demonstrates limited permeability for lipophilic antifungal agents [4]. A human nail is composed of the nail plate (thin structure of approximately 25 layers of keratinocytes with keratin filaments matrix) and four epithelial tissues: nail matrix, hyponychium, nail bed, and perionychium [5]. The physical characteristics of the nail plate depend on the types of keratin in the different nail layers. Keratin fibers are bound by cysteine-enriched proteins. The arrangement of keratin fibers and the stability of the proteins make the nail plate a low-permeable, solid structure. The superficial layers of the nail plate are relatively rich in phospholipids, but their total content in the nail is less than 1% (e.g., 10% in the *stratum corneum* of the skin). The amount of moisture in the nail depends on several factors. Usually, there is about 18% of moisture in the nail plate. If this amount drops to 16%, nails begin to break. Nails become extremely soft if moisture content increases up to 25%. It has been determined that the nail does not take more than 25% of moisture, and this effect often complicates the modeling of nail penetration—high permeation values require the highest possible moisture level in the nail. Drug substances could reach deeper layers of the nail if the disulfide bonds between the keratin molecules are broken [6].

Once the dermatophyte attaches to the surface of the nail, the filament-forming fungus starts to penetrate inside the nail plate, and the fungal cells begin to produce keratinocyte proteases that occupy the volume of healthy cells. Keratinocytic proteases promote the degradation of keratin to oligopeptides and amino acids [7]. Dermatophytes are unable to reduce the disulfide bonds that bind keratin molecules, but they weaken or partially deform them. Such an effect was found to depend on the individual sulfur content encoded in the TsuSsu1 genes [8]. Dermatophytes have parasitic properties—they multiply extremely rapidly in keratin-containing structures, but their eradication is a very complex and long-lasting process. The nail plate becomes opaque, loses its shine, and becomes brittle. The nail begins to turn yellow, thickens, is uneven, and rough [9,10]. The significant differences in healthy and infected nails are observed between the form and intensity of disulfide bonds [7]. In the infected nail, the disulfide bonds between the cysteine molecules have been elucidated: the strength and overlap of the S-S bond changes, resulting in changes of the mechanical properties of the nail. The nails of onychomycosis patients are prone to hyperkeratosis [11], their physicochemical properties change, and the nails become approximately three times thicker [12]. The complex structure of the nail plate and changes during pathology result in an extremely low penetration of drugs from topical preparations into the nail bed. The molecule, which is transferred through the nail plate, should have a low molecular weight and remain undissociated. Published scientific data indicate that the permeability of drug substances through the nail is promoted by the hydrophilicity of the drug substance, pH, the nature of the carrier, and the surface charge.

To treat nail diseases exclusively with topical preparations, it is necessary to guarantee that a higher than a minimal inhibitory concentration of drug substance will be established within the layer of hard keratin in the nail plate, and will be able to reach deeper layers. This could be achieved by applying physical, chemical, or mechanical methods to activate nail penetration [13]. Depending on their mechanism, they may cause different effects. The formation of internal cavities in the nail structure can be caused by low-frequency ultrasound. The phenomenon of cavitation (formation of air bubbles and sudden bursting) is observed in the penetration studies of active substances from liquid dosage forms. The sudden bursting of air bubbles in a liquid medium causes shock waves, and their passage through the biological membrane. This effect creates asymmetric pressure, causes the formation of internal cavities in the biological membrane [14], and statistically significantly increases the penetration of active substances into the nail membrane [15]. The application of fractional CO_2_ laser causes irreversible changes in the nail structure: formed microchannels increase the contact surface area between the nail and the formulation, resulting in greater permeability of drug substances through the nail barrier [16]. This effect was evaluated in clinical practice while treating onychomycosis: nails were pre-treated with CO_2_ laser radiation before the application of amorolfine cream: 22 patients out 24 showed a positive response, and 12 patients were completely cured of onychomycosis [17].

Permeability of the nail plate could be improved by applying electric field for the delivery of charged molecules. Negative charge iontophoresis has statistically significantly improved the transfer of mannitol and urea to the nail [18]. Nair AB et al. [19] have determined that iontophoresis could improve the transport of positively charged terbinafine hydrochloride molecules to the human nail, compared to passive transport. It has been observed that, during passive diffusion, the drug accumulated in the superficial layers of the nail, and after application of iontophoresis, terbinafine was detected in all layers of the nail plate. Additionally, it has been demonstrated that the combination of iontophoresis with surfactants improves the penetration of high molecular weight, water-insoluble molecules into the nail layers [20].

Alternatively, the application of chemical penetration enhancers results in irreversible changes in the natural nail structure. Sulfhydryl-containing substances cause swelling and increased porosity of the nail plate by disrupting disulfide bridges between keratin molecules, e.g., the corrosive effect of thioglycolic acid is irreversible [11]. Keratolytic substances (e.g., urea or salicylic acid) cause denaturation of keratin molecules, leading to the softening of the nail plate and pore formation in the nail structure. Urea can denature nail keratin by breaking the hydrogen bonds between keratin molecules [21].

The hydrophilicity of the nail structure determines that many transdermal enhancers are not applicable for improving drug substance penetration into the nail, as they act by affecting the lipids in the membranes [13]. The enhancement of transungual delivery is efficient when the chemical and physical structural integrity of the keratin structure is disrupted, affecting the main target molecules: disulfides, peptides, hydrogen, and polar bonds [22].

The penetration of drug substances into the nail layers is strongly influenced by used carrier systems. Due to the created reservoir of the drug substance in the nail and high concentration gradient, nail lacquers are one of the most perspective pharmaceutical dosage forms for application on the nail surface [23]. A high diffusion gradient is guaranteed for the permeability of the drug substance to the nail [24], and the formation of a film reduces the water loss from the nail structure, thus causing a hyperhydration of the superficial nail layer, which has a positive effect on the migration of drug substance [25].

In the current research, the dermatophyte growth inhibitor naftifine, [26] was selected as a model drug, because of its strong lipophilic characteristics. In medicinal practice, naftifine is applied for the treatment of fungal skin diseases in two different dosage forms—semi-solid formulation and skin solution. Previous studies have demonstrated that naftifine could inhibit dermatophytes, which are the main cause of onychomycosis [27,28]. MIC concentration of naftifine is in the range of 0.031–0.1 µg/mL, and its efficacy directly depends on used concentration [27,29]. However, there is no published data on possibilities to apply naftifine for transungual delivery. Additionally, in this research, naftifine containing nail lacquers were developed for the first time.

## 2. Results

### 2.1. Evaluation of Chemical Enhancers for Transungual Delivery of Naftifine

The excipients of the lacquers and their quantitative composition can have a crucial effect on drug release from the dried film of lacquer and its penetration into the nail. General procedures, described by Murthy et al. [30], were applied in screening for potential transungual chemical enhancers.

The selection of chemical enhancers was performed by evaluating the accumulation of naftifine hydrochloride in bovine hooves membranes that were used as a nail model. Monti et al. [31] validated bovine hoof membranes as a model for infected human toenails, and this nail model is widely used in penetration studies, especially in the case of the limited availability of human nail plates [32,33]. Compared to the control group (treated with acidified water pH 3), the penetration of naftifine into the nail was statistically significantly (*p* < 0.05) improved by the simultaneous use of 0.5% salicylic acid, 5% and 10% thioglycolic acid, 5% and 10% urea for enhancement of transungual penetration (Figure 1).

It was determined that low molecular weight polyethylene glycol 400 improved naftifine hydrochloride accumulation in the hoof, and this effect can be attributed to their ability to cause nail swelling. However, during pre-formulation studies, polyethylene glycol 400 was immiscible with components of the nail lacquers. Organic solvents and inorganic salts, irrespective of their concentration (5–50%) did not affect effect on the accumulation of an active substance in the nail.

A total of 10% thioglycolic acid affected the structure of the nail membrane after 24 h: corrosive effect (membrane color changes, visible cracks appeared), and loss of nail weight were determined (Figure 2). This effect is associated with the ability of thiols to disrupt disulfide bridges linking keratin molecules [18], which are responsible for the strength of the nail structure.

It was determined that 10% urea solutions have changed the shape and properties of the nail membranes—the nail surface has become uneven, wavy on the sides, which may be associated with the softening effect of the keratolytic agent.

To prevent nail membrane breakage, the solutions containing 5% thioglycolic acid, 5% urea, and 0.5% salicylic acid were used in the studies.

### 2.2. Impact of Chemical Enhancers on Experimental Naftifine Nail Lacquers Quality and Biopharmaceutical Characteristics

During the initial formulation phase, it was determined that urea was incompatible with the polymer: a white precipitate formed, and lacquer was not forming a film. Experimental compositions of nail lacquers with 5% thioglycolic acid and 0.5% salicylic acid were developed (Table 1).

It was determined that the addition of chemical enhancers did not affect nail lacquers’ qualitative characteristics. In all cases flowy and efficiently drying formulation was produced. The films stayed attached to the testing surface up to 5 days, regardless of used enhancers, and film was formed within 1 min.

The in vitro release profiles of naftifine from lacquer films were differently affected by the added chemical enhancers (Table 2). It was determined, that thioglycolic acid produced no statistically significant (*p* > 0.05) effect on the release of naftifine hydrochloride from nail lacquer formulations in 6 h. The difference was determined at the beginning of the study—thioglycolic acid addition caused a burst effect on naftifine release during the first two hours of the study. The specific odor of the film containing thioglycolic acid was identified.

A decrease on naftifine release was determined when salicylic acid (as enhancer) was added to the formulations. The release of the naftifine decreased statistically significantly (*p* < 0.05) if compared to enhancer-free (control) formulation. The presence of salicylic acid caused the prolonged release of naftifine hydrochloride: after 6 h the amount of drug substance (87 ± 0.78%) was lower than in the control group (96 ± 0.64%). This effect can be attributed to the possible interaction between drug, polymer, and enhancer. Regarding the results of previous studies, all three formulations were selected for further transungual penetration testing studies.

### 2.3. Transungual Delivery of Naftifine

Transungual penetration studies of naftifine were performed using non-laser treated and laser pretreated human nail clippings, and non-laser treated bovine hooves. In this study, we used healthy volunteers nail clippings. The use of the nail clippings from the same few volunteers guaranteed a controlled variation of nail thickness, while it was demonstrated that drug permeability could be considerably influenced by the nail thickness [34]. Three nail lacquer formulations (N1, N2, and N3) were tested, and the results of penetration were evaluated.

The influence of chemical enhancers on the penetration of naftifine hydrochloride was assessed using two different nail models. The evaluation of the accumulation of naftifine hydrochloride and its penetration in bovine hooves indicated that the addition of chemical agents had no statistically significant (*p* > 0.05) effect on the accumulation of naftifine hydrochloride in hoof layers.

Addition of thioglycolic acid and salicylic acid statistically significantly (*p* < 0.05) improved the permeation of naftifine through the hoof membrane: 14 ± 1.9%, and 14 ± 0.4%, respectively (Figure 3).

Opposing results were attained when human nail clippings were used. It was determined that naftifine is less permeable through human nail compared to other antifungal agents, e.g., amorolfine [35]. The highest amount of naftifine hydrochloride (15 ± 0.4%) accumulated in the nail, when thioglycolic acid was included in the formulation. Transungual penetration can be affected by the physicochemical properties of the drug molecule. It has been determined that smaller molecules diffuse better through the pores of the keratin network [36]. Naftifine should penetrate relatively easy into the nail plate, as its molecular weight of 32,386 g/mol is considered medium, and is even lower than one of the most frequently used antifungal agents—amorolfine (353,975 g/mol). However, penetration results are lower and this may be associated with a high lipophilicity of naftifine hydrochloride—since the nail plate behaves like a highly concentrated hydrophilic gel [37], therefore, it is problematic for the drug molecule to diffuse through the nail membrane and low pH. According to A. Gupta et al. [38], the binding of naftifine to membranes are dependent on pH values. Low naftifine efficacy corresponds to low binding, due to decreased pH. However, naftifine solubility is higher at pH 3. The results demonstrated that neither thioglycolic acid nor salicylic acid had a statistically significant (*p* > 0.05) effect on the penetration of naftifine hydrochloride through the human nail—3–6% of drug substance was detected in the acceptor phase. Comparing both used nail models, it was observed, that the total accumulated and penetrated amount of naftifine were similar. However, it can be concluded that bovine hooves are more permeable than human nails, and this model is not fully representative to mimic the effect of chemical enhancers on the transungual delivery of antifungals in the human nail.

Physical enhancement is an alternative approach to increase antifungal agents’ penetration into the nail. Two different methods were chosen for further testing: (i) ultrasound, as a minimally invasive method [14] and opposite to that; (ii) fractionated CO_2_ laser, as a clinically approved method [39]. The effect of physical enhancers on transungual permeation of naftifine hydrochloride was determined by using human nail clippings (Figure 4). The application of fractionated CO_2_ laser in “fusion” mode has statistically significantly (*p* < 0.05) increased the accumulation of naftifine hydrochloride (10 ± 0.5%), compared to the control (7 ± 0.8%) in the nail.

Combining this method with ultrasound, a statistically significant (*p* < 0.05) improvement in naftifine permeability (11 ± 0.5%) was determined. This result could be related to the ultrasound’s cavitation effect [15] when micrometer range bubbles in liquid are formed and increases diffusion of drug substance by disrupting collagen structure. The use of a “deep” laser application mode did not result in a statistically significant effect on (*p* > 0.05) accumulation and penetration of naftifine hydrochloride. This result demonstrates that enhancement effect is not directly related to the applied laser power. Nail surface changes induced by physical and chemical pretreatment enhancement are presented in Figure 5.

The simultaneous effect of laser pretreatment and the presence of chemical enhancers in the lacquer on naftifine permeation was established. It was determined that the combination of chemical and physical enhancers caused a statistically significant (*p* < 0.05) increase of naftifine penetration (Figure 6).

The highest amount of naftifine (25 ± 0.6%) accumulated in the nail layers after the combined effect of “fusion” laser and thioglycolic acid. The combination of the “fusion” mode of laser application and thioglycolic acid resulted in a statistically significant (*p* < 0.05) increase in the penetration of naftifine hydrochloride through the nail and increased amount of the drug substance in the acceptor medium. Salicylic acid also produced a statistically significant (*p* < 0.05) effect on the accumulation of naftifine hydrochloride when combined with “fusion” mode of laser application. However, the combination of these enhancers did not affect the permeability of the naftifine hydrochloride through the nail.

Summarizing the results of transungual permeation studies, it has to be emphasized that the determined concentrations of naftifine in the tested biological membranes and in the acceptor phase exceeded minimal inhibitory concentration (MIC) levels [27]. Thus, it could be expected, that naftifine should be efficient in onychomycosis treatment, in combination with appropriately applied chemical and physical penetration enhancement techniques. Although knowledge about nail permeability and transungual delivery has increased recently, the effects of microstructure of the nail plate, drug-keratin binding properties, or structural models for predicting nail delivery are still not fully clear [4]. The results presented here provides new knowledge on the transungual penetration of naftifine and factors, affecting these processes.

## 3. Materials and Methods

### 3.1. Materials

Naftifine hydrochloride was purchased from ChemicalPoint (Deisenhofer, Germany). Ethanol 96% (Vilniaus degtinė, Vilnius, Lithuania), butyl acetate and ethyl acetate were obtained from Sigma-Aldrich Chemie GmbH (Steinheim, Germany) and used as a solvent system. Triacetin which was used as plasticizer was kindly supplied by Lanxess (Leverkusen, Germany). Film-forming polymer Eudragit RL100 was kindly gifted by Evonik Industries AG (Essen, Germany). Salicylic acid (Alfa Aesar, Karlsruhe, Germany), glycerol (Applichem, Darmstad, Germany); 1.2-propandiol, polyethylenglycol 400, polyethylenglycol 1500, urea was purchased from Roth (Karlsruhe, Germany) and used as chemical enhancers. Tween 60, Tween 40, citric acid monohydrate, sodium carbonate, acetone, benzoic acid, and methanol were used as enhancers and obtained from Sigma-Aldrich Chemie GmbH (Steinheim, Germany). Thioglycolic acid was kindly gifted by Merck Group (Darmstadt, Germany). Acetonitrile and trifluoracetic acid for chromatography analysis were purchased from Sigma-Aldrich Chemie GmbH (Steinheim, Germany).

### 3.2. Methods

#### 3.2.1. Preparation of Hooves and Nails

Hooves were taken from freshly slaughtered 2–3-year-old cattle, stripped of adhering cartilaginous, connective tissue and kept in distilled water for 72 h [40]. Sections, approximately 60 µm thick, were taken from the bottom of the hoof with cryotome (Thermo Scientific Cryotome FSE, Cheshire, UK). Prepared hoof membranes were kept at −20 °C and moved to room temperature 5 h before use in penetration studies.

Nail clippings were obtained from healthy human volunteers (male and female, age 25–55 years) using nail clippers. Nail clippings were washed with phosphate buffer (pH 7.4) and wiped with filter paper.

#### 3.2.2. Selection and Evaluation of Chemical Enhancers

The predefined quantities of enhancers (listed in Figure 1) were included in acidified water and the final pH was adjusted to 3, to achieve necessary maximum solubility of naftifine hydrochloride. The concentrations of applied enhancers were chosen referring to their solubility at pH 3, as per the previously published scientific data [35].

Bovine hoof membranes were placed into the above solutions and kept at 32 °C for 24 h. After washing and drying, hoof membranes were weighed, placed into naftifine hydrochloride solution (500 µg/mL) and incubated for 24 h at 35 °C. Membranes were washed and active substance was extracted with methanol by sonication for 30 min and analyzed for naftifine hydrochloride content. The controls were carried out using a “drug only” formulation containing 500 µg/mL naftifine hydrochloride solution in acidified (pH 3) water.

#### 3.2.3. Physical Enhancement

The evaluation of physical enhancers was performed by using 800 µm human nail clippings as a nail model system. Fractional CO_2_ laser and ultrasound were applied as efficient physical enhancement methods.

Candela CO_2_RE laser (Syneron Candela, Wayland, MA, USA) was used to disrupt the nail barrier to enhance transungual permeation of naftifine hydrochloride. Nail clippings were treated by the laser at different energy levels: “fusion” (50–70 mJ energy) and “deep” (60–80 mJ energy) before the application of nail lacquers.

To determine the possible effect of ultrasound on transungual permeation of naftifine, nail clippings were placed in ultrasonic bath USC 1200 THD (VWR, Penang, Malaysia) for 15 min, 30 min, and 45 min. Ultrasound was used before or after nail lacquer application.

After using both methods separately, ultrasound and “fusion” mode energy level laser were concomitantly applied on nails before nail lacquer application. Nail clippings were treated by laser and placed in an ultrasonic bath for 30 min. After these steps, nail lacquer was applied and left until the “dry-to-touch” condition was achieved.

Structural changes of the nail were visualized by optical microscope Optika B-353FL (Optika, Ponteranica, Italy).

#### 3.2.4. Preparation and Evaluation of Experimental Formulations

The experimental nail lacquers (Table 2) were formulated by dissolving the amount of naftifine in a lacquer base, containing solvent mixture of ethanol, ethyl acetate and butyl acetate, plasticizer triacetin, film-forming polymer Eudragit RL100, and an appropriate chemical enhancer. The composition of the enhancer-free formulation was optimized in a previous study [29].

Drying time was evaluated by applying a liquid film of the experimental sample on a glass plate and the time until obtaining a dry-to-touch state was determined. The water resistance test was performed by applying a nail lacquer onto a glass slide, allowing it to dry, then immersing it into distilled water for 7 days [41,42].

Naftifine hydrochloride release experiments were carried out at 32 °C, measuring drug quantity diffusing through 1 cm^2^ cuprophan dialysis membranes (MWCO—10,000 Da). A diffusion membrane was mounted on the diffusion cell, and 50 µL of lacquer experimental formulation was applied uniformly on the surface of the membrane and left for 4 h, until complete lacquer drying and film formation. Dry film was protected from external factors by aluminum foil cover. Acidified water (pH 3) was used for naftifine hydrochloride release testing, and this solvent was confirmed as establishing *sink* conditions. At predefined time points, 1 mL of acceptor medium was withdrawn, and the same amount of a fresh medium was added to maintain a constant volume of acceptor medium. The released amount of naftifine was determined by UPLC chromatography in triplicates.

#### 3.2.5. Nail Penetration Studies

Naftifine hydrochloride transungual permeation was studied by determining drug substance quantity in human nail clippings and acceptor phase. The lacquer sample was applied in three layers, allowing complete drying until constant weight. Transungual delivery studies were performed by applying “wetted cotton ball method” [43] for 24 h at 32 °C: nail clippings were placed on small wetted cotton balls, which functioned as an acceptor compartment and provided moisture to the nail plate. The average surface area of nail clippings was 0.5 cm^2^. The “wetted cotton ball method” was chosen, due to the fact that the minimum *Franz* cells acceptor phase volume was too big for nail clippings penetration studies. It was determined that drug substance cannot be detected in the acceptor phase samples during penetration studies. By using the “wetted cotton ball method”, reproducible experimental conditions were created that reproduced the nail bed.

Afterward, the nails were cleaned, washed, and dried. The weight and thickness of each nail were measured. All clippings were transferred to Eppendorf tubes and extracted with 1 mL of methanol at 32 °C for 24 h. Naftifine hydrochloride in cotton balls was determined by extracting with methanol for 24 h at 32 °C. The amount of naftifine was determined by UPLC chromatography in triplicates. Transungual penetration studies were approved by Kaunas Region Bioethical Committee (corresponding bioethical permission approval number BE-2-41).

#### 3.2.6. Analytical Procedures

Naftifine content was quantified by liquid chromatography using Acquity UPLC H-Class chromatography system (Waters, Milford, MA, USA) equipped with DAD (Waters, Milford, MA, USA), performing detection at 294 nm. Separation was performed on Acquity UPLC BEH C18 (130 Å, 1.7 µm, 2.1 mm × 50 mm, Waters, Milford, MA, USA) column. The mobile phase was delivered in a linear elution gradient from 70% to 30% of solvent A (acetonitrile) in B (0.1% (*v*/*v*) trifluoracetic acid in ultrapure water) for 5 min; the injection volume was 1 µL, flow rate was 0.7 mL/min, and the column temperature was 30 °C. A standard calibration curve was built up by using standard solutions (0.5–121.5 µg/mL). The developed UPLC method was validated in terms of precision, accuracy and linearity according to ICH guidelines [44]. Assay method precision was carried at LOQ level and determined using seven independent test solutions. The %RSD was within the acceptable limit of 2%. Linear calibration plots were obtained at seven concentration levels in triplicate and linearity confirmed for the range 0.5–121.5 µg/mL (R^2^ = 0.9999). The results showed excellent correlation between the peak area and concentration. The accuracy of the assay method was evaluated with the recovery of the standards from samples in triplicate. The recovery of the investigated components ranged from 98.8% to 100.1%, and their RSD values were less than 3%, characterizing good reliability and accuracy of the method. The LOD for naftifine was 0.0131 µg/mL and LOQ was 0.0437 µg/mL.

#### 3.2.7. Statistical Analysis

The results are presented as means ± SD. Spearman‘s rank coefficient was used for correlation analysis. Statistically significant difference was determined when value of *p* < 0.05. The results were calculated using IBM SPSS Statistics for Windows Version 19.0 (IBM Corporation, Armonk, NY, USA) and Microsoft Office Excel 2015.

## 4. Conclusions

The current study demonstrated, that naftifine is accumulating in the nail, however, a small fraction of the applied dose penetrates through the nail barrier. The results of the study confirm the importance of the appropriate combination of the applied enhancement methods, to achieve increased accumulation and penetration rates of the drug substance. Applied physical enhancement methods can be considered an alternative tool that can affect nail barrier properties and increase the efficacy of the treatment. The application of low-frequency ultrasound does not cause pain and has a reversible effect. The fractional CO_2_ laser, which is widely used in cosmetology practice, causes mechanical damage to the nail structure, which results in formation of microchannels and increased surface area for penetration. When this method is combined with the corrosive effect of thioglycolic acid, increased penetration of naftifine into the inner layers of the nail can be expected. These findings should be validated with additional in vitro and ex vivo experiments.

However, it may be too early to conclude that naftifine could be used in clinical practice for onychomycosis treatment. From the current perspective, there are several antifungal agents that demonstrate more efficient penetration in comparison to naftifine, without the application of enhancement techniques. However, naftifine offers its advantages, as there is no data about naftifine attachment to nail keratin, meaning that the application of lower naftifine concentration products can produce therapeutically efficient concentrations of the drug substance. Secondly, considering the fact that onychomycosis topical treatment can last a few months, the evaluation of possible alternative treatment options for long term therapies could be considered a promising and practically valuable research topic.

## Figures and Tables

**Figure 1 molecules-25-03043-f001:**
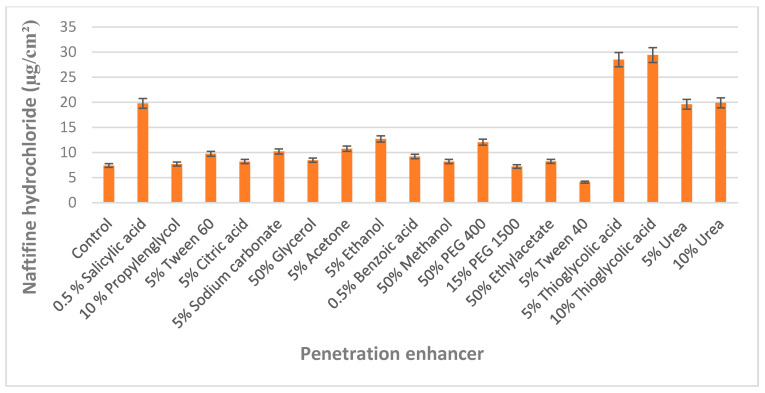
Effect of chemical enhancers on the accumulation of naftifine in the bovine hoof.

**Figure 2 molecules-25-03043-f002:**
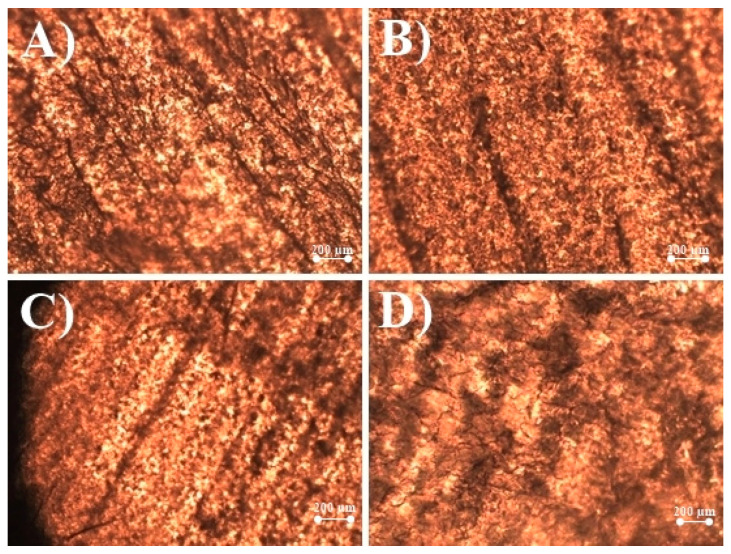
Human nail clippings surface changes after chemical enhancers’ application: (**A**) natural structure human nail surface; (**B**) human nail after 24 h in 5% urea solution; (**C**) human nail after 24 h in 0.5% salicylic acid solution; (**D**) human nail after 24 h in 5% thioglycolic acid solution.

**Figure 3 molecules-25-03043-f003:**
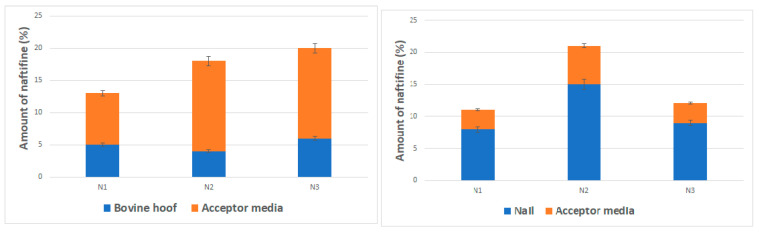
Penetration of naftifine in different nail models. N1—enhancer-free nail lacquer formulation. N2—5% thioglycolic acid nail lacquer formulation. N3—0.5% salicylic acid nail lacquer formulation.

**Figure 4 molecules-25-03043-f004:**
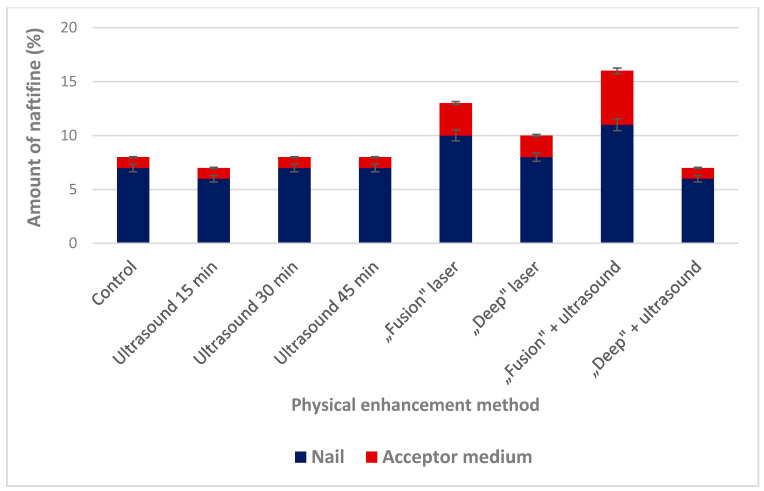
Effect of physical enhancers on transungual permeation of naftifine.

**Figure 5 molecules-25-03043-f005:**
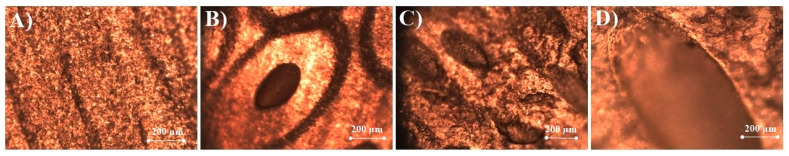
Effect of different enhancement methods on the human nail surface. (**A**) nail surface after 30 min ultrasound application; (**B**) laser affected nail surface; (**C**) laser-affected nail surface after 24 h application of nail lacquer with 5% thioglycolic acid as chemical enhancer. (**D**) laser affected nail surface after 30 min application of ultrasound.

**Figure 6 molecules-25-03043-f006:**
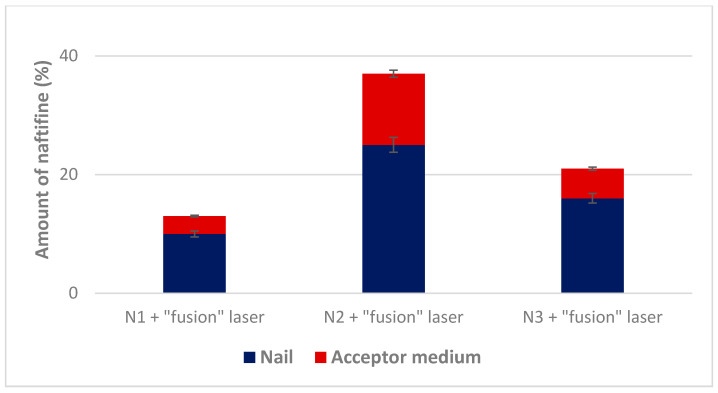
Effect of chemical enhancers combined with CO_2_ fusion laser pre-treatment on naftifine nail permeation.

**Table 1 molecules-25-03043-t001:** Compositions of formulated experimental lacquers with naftifine.

Formulation Code	N1	N2	N3
Eudragit RL100	15	15	15
Solvent mixture (1:1:1)	79	74	78.5
Triacetin	5	5	5
Thioglycolic acid	-	5	-
Salicylic acid	-	-	0.5
Naftifine hydrochloride	5	5	5

**Table 2 molecules-25-03043-t002:** Naftifine hydrochloride release (%) from experimental lacquers.

Time (h)		Release Amount (%)					
		1	2	3	4	5	6
Formulation code	N1	24 ± 0.2	45 ± 0.13	63 ± 0.3	84 ± 0.4	90 ± 0.2	96 ± 0.32
	N2	45 ± 0.1	59 ± 0.4	76 ± 0.42	89 ± 0.53	91 ± 0.56	97 ± 0.46
	N3	28 ± 0.21	39 ± 0.5	52 ± 0.64	67 ± 0.4	75 ± 0.76	87 ± 0.3

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
