# Peer review of "Potential of Naftifine Application for Transungual Delivery"

_molecules, 2020, doi:10.3390/molecules25133043_

Round 1

Reviewer 1 Report

This manuscript describes the ability of naftifine to accumulate in the nail layers and penetrate through the human nail barrier. Naftifine is known for its antifungal activity however, it is not used in onychomycosis treatment.

The main advantage of this article is novelty of the research. The authors produced naftifine polymeric nail lacquer and applied naftifine for transungual delivery for the first time. For better nail penetration some chemical and physical enhancing factors were applied. In my opinion, the results obtained are not impressive. Although the authors showed increased accumulation of naftifine with thioglycolic acid in the nail layers by 100% compared with a control group, its level was still relatively low, amounting to 15%. Moreover, the results demonstrated that neither thioglycolic acid nor salicylic acid had a statistically significant influence on the penetration of naftifine through the human nail. The best results were obtained when thioglycolic acid-containing formulation was applied and the nail was pretreated with a fractional CO2 laser.

In the Introduction the authors write; Barrier changes in pathology become an additional challenge in permeation studies-the human nail is one of the natural barriers that changes during infections and creates additional challenges for efficient therapy.

In my opinion, research should be extended to include a model with an infected human nails from patients or at least a simulated nail infection in the laboratory by one of dermatophyte species.

The conclusion section is very short and not well prepared and it mainly repeats the results.

How do the authors see the simultaneous application of lacquer and CO2 laser in the medical treatment? Would such a solution be practical for the patient? Does naftifine really have reasonable potential to be commonly used to treat onychomycosis? How does naftifine compare to other medicines (in lacquer form) used for onychomycosis treatment?

The major concern of this paper is the incorrect placement of tables and not very detailed descriptions of the figures, which makes it difficult for the reader to follow the logical sense of the results.

For example, in Table 1  and Figure 2 on page 4, the authors use the designations N1-N3, which were not previously explained in the text or in the descriptions of the Table/Figure, and then the reader reads; Decrease on naftifine release was determined when salicylic acid was added to the formulations. And how can the reader know which results to look at in the table and in which formulation (N1-N3) was the salicylic acid? Only at the end of the paper, in the methods section, the reader learns what N1-N3 means. In my opinion, Table 2 on page 8, should appear in the description of results before Table 1 or in the description of the Table1/Figure2 on page 4, should appear information what N1-N3 means.

Figure 5 on page 7 should be moved to the results or supplementary materials.

In the Abstract the authors write; Structural changes of nail barrier, induced by fractional CO2 laser, were visualized by microscopy. I do not see any picture of this observation at least in my version of the file and there is no information in Methods how was it visualized also in case of Figure 1 on page 3. In the description of this Figure should be added if it is human nail of bovine hoof.

In the Method section there is no information on how the authors used ultrasounds and laser together. What was used first, at what time etc.

In all experiments the information about the number of repetitions is missing and it is the key information for statistical analysis.

Additional minor items

line 16. what is „t naftifine” ?

line 22. and others CO2 should be CO2

line 35. 50% all cases – “of” should be added

line 42. “which is also demonstrates limited permeability” should be corrected

line 44. “ant” should be corrected

line 73. sentence “In current research, the growth of dermatophytes inhibiting naftifine” should be corrected

line 99. (Fig. 1) should be at the end of the sentence

line 105-106. There should be a space between 24 h

line 119. Table 1 should be at the end of the sentence

line 149. then should be changed to when

To sum up, this manuscript deserves to be published in Molecules after major revision.

Reviewer 2 Report

The authors need to improvise this manuscript significantly. The introduction is poorly written and has lack of understanding of nail structure.

Specially, hydration of nail plate has limited information.

Authors should consider about partial and fully hydrated nail plate and mention the fiber volume fraction of the nail and relative permeability of charged and uncharged solutes.

How did results come before the method??? It's a basic 1.01 of writing manuscript.

The discussion is poorly written and must be improvised.

Is wetted cotton ball method a new method? Who else has used this method in the literature?? Why did authors not use a established franz cell method with nail adapters?

Reviewer 3 Report

Naftifine is an allylamine developed in the 1970-1980s. It did not gain much success as its successor drug, terbinafine, was thought to be more promising. So, only a cream containing naftifine was marketed whereas terbinafine was developed for oral and topical use. Despite its excellent MiC values several experimental terbinafine lacquers were not good enough to be marketed. The authors now tested penetration enhancers – thioglycolate and laser nail plate penetration – on human nail clippings. Urea was found to be incompatible with a film-form solution (which is surprising as there are several urea containing nail lacquers available). Naftifine release from lacquer films was not influenced by any enhancer – however, table 1 does not give any information as which N formulation contains what. Again, in the nail penetration study, the formulations of the lacquers are not given, which is a considerable disadvantage.

Line 128, 132: What do you mean by enhancer?

Line 163: “fraction CO2 laser” is fractionated CO2 laser.

Line 165: What is a “fusion laser”?

Conclusion: This is a very bold statement that naftifine may be used in a lacquer  base as a treatment for onychomycosis. Laser pretreatment with fractionated CO2 laser is not very practical and would make the therapy excessively expensive. The incorporation of thioglycolate into a nail lacquer is not new and was actually tested about 30 years ago, but this approach was not followed then. On the other hand, it is justified to test a drug already known as being safe and active.

Round 2

Reviewer 1 Report

In my opinion, the manuscript now deserves to be published in Molecules. The authors introduced extensive corrections in the Introduction and Conclusions section suggested by the reviewers. Also the arrangement of figures and tables now has a logical order and proper descriptions. I just think that Table 3 in the Method section should be removed because it already appears as Table 1 in the Results. I would recommend that the authors re-analyze the text from the editorial side because there are some minor language errors, e.g. line 370 "is can", incorrect punctuation or missing spaces (e.g. line 11). I also think that the font on some graphs is too big (e.g. Fig. 4 & Fig. 6). In summary, after a language correction, I recommend this paper to be published in Molecules.

Author Response

Point. In my opinion, the manuscript now deserves to be published in Molecules. The authors introduced extensive corrections in the Introduction and Conclusions section suggested by the reviewers. Also the arrangement of figures and tables now has a logical order and proper descriptions. I just think that Table 3 in the Method section should be removed because it already appears as Table 1 in the Results. I would recommend that the authors re-analyze the text from the editorial side because there are some minor language errors, e.g. line 370 "is can", incorrect punctuation or missing spaces (e.g. line 11). I also think that the font on some graphs is too big (e.g. Fig. 4 & Fig. 6). In summary, after a language correction, I recommend this paper to be published in Molecules.

Response. Authors are grateful for all comments, ideas and questions. We had revised manuscript, corrected grammar, spelling, style and punctuation errors. Also, Table 3 was removed as it was used second time in manuscript.

Reviewer 2 Report

The authors needs to further revise this manuscript. 

Major comments:

1. The authors did not mention the impact of isoelectric point/pH on permeation characteristics or diffusion of Naftifine in regards to the charged state of the molecule? Some cues can be taken from –

Smith et al. 2009. Effects of ionic strength on passive……

Baswan et al. 2012 Characterization of ion transport……

Baswan et al. 2016 Size and charge dependence of ion transport…..

2. Authors should further improve the discussion on the hypothesis/factors influencing penetration from the active molecule standpoint. Influence of MW is more predominant than the log P/lipophilicity of the molecule. This was supported by a comprehensive dataset comparison of 42 solutes in the nail plate.

Baswan et al. 2016. Diffusion of uncharged solutes through….

3. The authors should improve the conclusion by stating the need for more invitro and invivo studies to validate results.

4. While discussing the cotton ball method, the author should discuss it in reference to the disadvantages of franz cell adpapter methods referring to Edge effects.

5. Edge effects: As authors use a very small surface area for diffusion/penetration studies, there is something called “edge effects” which can introduce systemic errors in the dataset due to lateral diffusion. The following references discuss this at great length. The authors should mention this caveat or limitation of their study and discuss it in detail. The influence of lateral diffusion in such cases can be significant and dependent upon the total size of the membrane relative to the exposed surface.

Palliyil et al. 2014 Lateral drug diffusion in human nails.

Barrer, et al. 1962  Permeation through a membrane with mixed boundary conditions.

Baswan 2014. Transport of Charged and Uncharged Solutes…….

6. Authors must improve upon all the figures. The background color and data is presented in a naive style. Especially, no attention has been given to labeling X axis of Figure 4.  

7. The english language must be improved as a whole in the manuscript.

8. The conclusion could be condensed to give an impactful message.

Author Response

Point.1. The authors did not mention the impact of isoelectric point/pH on permeation characteristics or diffusion of Naftifine in regards to the charged state of the molecule?

Response 1. We agree that formulation with isoelectric point/pH could affect the release and penetration results. In this study we focused on formulating stable lacquers supporting solubility of drug substance. There is no scientific data published (to our knowledge) on the effect of charged state of naftifine on its transungual permeation. Some further research with iontophoretic enhancement were just initiated in our lab.

Point 2. Authors should further improve the discussion on the hypothesis/factors influencing penetration from the active molecule standpoint. Influence of MW is more predominant than the log P/lipophilicity of the molecule. This was supported by a comprehensive dataset comparison of 42 solutes in the nail plate.

Response 2. According to available published data, naftifine has even lower MW than amorolfine (amorolfine is used for onychomycosis treatment and is one of the most promising antifungal agents), which means, that naftifine should penetrate better into the nail plate, if compared to amorolfine. By comparing these two active substances and their penetration, we may conclude, that MW of naftifine is not that predominant factor (the volume of the drug molecule could be considered as alternative).

Point 3. The authors should improve the conclusion by stating the need for more invitro and invivo studies to validate results.

Response 3. Appropriate modifications were made. Authors agree, that more ex vivo experiments would be make difference in methodology validation.

Point 4. While discussing the cotton ball method, the author should discuss it in reference to the disadvantages of franz cell adpapter methods referring to Edge effects. Edge effects: As authors use a very small surface area for diffusion/penetration studies, there is something called “edge effects” which can introduce systemic errors in the dataset due to lateral diffusion. The following references discuss this at great length. The authors should mention this caveat or limitation of their study and discuss it in detail. The influence of lateral diffusion in such cases can be significant and dependent upon the total size of the membrane relative to the exposed surface.

Response 4. “Wetted cotton ball” method was developed and validated by scientists from Martin Luther University. This methodology was intercepted from them during experiments in Martin Luther University and transferred to our lab. During experiments, no lateral diffusion was observed, as the surface area is not as small as used in Biji B. Palliyil et al. experiments. Lateral diffusion effect was observed when cadaver nails were used and 3 mm sections of nail plate was cut. In our experiments nail clippings of approx. 0,5 cm2 were used.

Point 5. Authors must improve upon all the figures. The background color and data is presented in a naive style. Especially, no attention has been given to labeling X axis of Figure 4.

Response 5. Graphs were improved, by labelling axes, changing font size, font and colours.

Point 6. The english language must be improved as a whole in the manuscript.

Response 6. English language was revised in whole manuscript: grammar, spelling, style and punctuation mistakes were corrected.

Point 7. The conclusion could be condensed to give an impactful message.

Response 7. The conclusions of manuscript was rewritten trying to consider the comments of all reviewers.

Round 3

Reviewer 2 Report

The authors have made relevant changes to the manuscript.